# The Diversity of the Mitochondrial Outer Membrane Protein Import Channels: Emerging Targets for Modulation

**DOI:** 10.3390/molecules26134087

**Published:** 2021-07-04

**Authors:** Monika Mazur, Hanna Kmita, Małgorzata Wojtkowska

**Affiliations:** Institute of Molecular Biology and Biotechnology, Faculty of Biology, Adam Mickiewicz University, 61-614 Poznan, Poland; monika.antoniewicz@gmail.com (M.M.); kmita@amu.edu.pl (H.K.)

**Keywords:** mitochondria, import channel, TOM, TOB/SAM, Tom40, Tob55/Sam50, Mdm10, MIM, MAC

## Abstract

The functioning of mitochondria and their biogenesis are largely based on the proper function of the mitochondrial outer membrane channels, which selectively recognise and import proteins but also transport a wide range of other molecules, including metabolites, inorganic ions and nucleic acids. To date, nine channels have been identified in the mitochondrial outer membrane of which at least half represent the mitochondrial protein import apparatus. When compared to the mitochondrial inner membrane, the presented channels are mostly constitutively open and consequently may participate in transport of different molecules and contribute to relevant changes in the outer membrane permeability based on the channel conductance. In this review, we focus on the channel structure, properties and transported molecules as well as aspects important to their modulation. This information could be used for future studies of the cellular processes mediated by these channels, mitochondrial functioning and therapies for mitochondria-linked diseases.

## 1. Introduction

According to the endosymbiotic theory, mitochondria arose from prokaryotes through (endo)symbiosis. These double-membrane organelles have retained the oxidative phosphorylation system to synthesize ATP. This is the best-known function of mitochondria, which makes them unique and crucial for cells. However, in the course of evolution, nearly all ancestral genes were transferred into the nucleus. As a result, most mitochondrial proteins are synthesized in the cytosol and must be delivered to their correct mitochondrial destination [1,2]. Thus, protein import into the mitochondria seems to be additional evidence for the single origin of these organelles.

Since the last years, studies of protein import into mitochondria have revealed many aspects concerning the composition and function of the protein import machinery, which consists of protein import complexes with the main module working as an import channel that allows protein transfer across and into both mitochondrial membranes. The first barrier for incoming proteins is the mitochondrial outer membrane, which has approximately 200 proteins, of which some display channel activity [3]. To date, nine different channel-forming proteins responsible for the transport of metabolites, inorganic ions and proteins have been distinguished in the outer membrane [4]. Some of these proteins have a β-barrel structure and include Tom40, Tob55/Sam50, the voltage-dependent anion-selective channel (VDAC) and Mdm10, mitochondrial apoptosis channel (MAC) while those with other types of structure include Mim1, Ayr1, OMC7 and OMC8 [5]. The protein channels described in this review, except of the MAC, are constitutively open and do not need activators to regulate its opening. This phenomenon makes them different than channels located within the inner mitochondria membrane.

In this review, we focus on the outer membrane protein import channels characterized as single modules and as subunits of the native complexes. We present current data about their structure, regulation and variety of transported molecules. Such knowledge is crucial for development of the channels’ modulators which could prove effective in correcting mitochondrial processes that in turn may constitute an important therapeutic approach in disease treatment.

## 2. Overview of the Mitochondrial Protein Import Machinery

The successful import of precursor proteins into their destinations in mitochondria primarily depends on the mitochondrial membrane complexes and specific signals localised within the incoming precursor proteins. The membrane import complexes—known as translocases—are composed of receptor subunits that recognise incoming precursor proteins, proteins that form channels for transport across membranes or insertion into membranes and protein subunits that modulate the activity and stability of complexes. In the mitochondrial outer membrane, four complexes have been identified (Figure 1): translocase of the outer mitochondrial membrane (TOM) [6,7], topogenesis of the mitochondrial outer membrane β-barrel proteins/sorting and assembly machinery (TOB/SAM) [8,9,10,11], mitochondrial import complex (MIM) [12,13] and endoplasmic reticulum-mitochondria encounter structure (ERMES) [14]. The TOM complex recognises, translocates and segregates most of the precursor proteins delivered to different mitochondrial locations. The TOB/SAM complex inserts into the outer membrane proteins of the β-barrel structure displayed by proteins forming channels, such as Tom40, Tob55/Sam50, mitochondrial distribution and morphology protein 10 (Mdm10; part of the ERMES complex) and mitochondrial porin (a VDAC responsible for metabolite and inorganic ion transport in and out of the mitochondria) [2]. The MIM complex inserts the α helical outer membrane proteins, which are imported independently of the TOM complex [12].

Precursor proteins destined for the intermembrane space (IMS) are assisted by the mitochondrial intermembrane space and assembly (MIA) complex [15]. Other precursors are transported or inserted into the inner membrane by two complexes (i.e., TIM23 [15,16] and TIM22 [17,18]). The TIM23 complex primarily performs import into the mitochondrial matrix, whereas the TIM22 complex performs insertion into the inner membranes of carrier proteins (for more details on the import across and into the inner membrane, see the following reviews: [19,20]) (Figure 1).

### Protein Import across and into the Mitochondrial Outer Membrane

The most widely used model organism in studies on mitochondria protein import machinery is the yeast *Saccharomyces cerevisiae*, which is also used as a reference for comparative studies on other eukaryotic organisms, including plants and animals [2,21]. As previously mentioned, the TOM complex receptors specifically recognise the incoming precursor proteins. Tom20 binds to the hydrophobic surface of the cleavable amphipathic presequence and initiates contact with the next receptor (i.e., Tom22 protein via its positively charged surface). In the case of non-cleavable hydrophobic precursors serving as carrier proteins with the inner membrane as the destination, the Tom70 receptor is engaged. Crossing of the outer membrane is performed by channel-forming Tom40 protein and small Tom subunits (e.g., Tom5, Tom6 and Tom7) that stabilize the complex and contribute to the proper assembly of its subunits [22]. From this entry point, import pathways diverge into different directions depending on the final destination of the protein [2] (Figure 1). After crossing through the channel formed by Tom40, proteins that adopted a β-barrel structure are bound to small TIM proteins in the IMS (i.e., Tim9 and Tim10) [23,24] and delivered to the TOB/SAM complex. The TOB/SAM complex consists of two peripheral membrane proteins exposed to the cytosol, namely Tob38 (also known as Sam35) and Tob37 (also known as Sam37/Mas37), as well as the channel-forming protein Tob55/Sam50 [25,26,27,28,29]. Thus, the TOB/SAM complex recognises β-barrel proteins and assists their membrane insertion and assembly [3,30]. Available data concerning the mammalian TOB/SAM complex indicate the presence of weakly conserved homologues of Tob37 and Tob38, known as metaxin 1 (Mtx1) and metaxin 2 (Mtx2), respectively [30,31,32]. The TOB/SAM complex of the slime mould *Dictyostelium discoideum* includes Tob55/Sam50 and one metaxin homolog [33,34,35]. It has also been proven that β-barrel proteins interact with the mitochondrial contact site and cristae organising system (MICOS) complex, which maintains the characteristic pleated structure of the mitochondrial inner membrane [36,37]. One of the MICOS complex subunits, Mic60/mitofilin, stimulates β-barrel protein transfer from the TOM complex to the TOB/SAM complex [38]. The outer membrane proteins anchored by one or more α-helical transmembrane segments are mostly inserted by the MIM complex, which consists of Mim1 and Mim2 protein subunits [4]. Additionally, the MIM complex cooperates with the protein Mdm10, which is part of the ERMES complex [14].

## 3. Structure and Properties of the Outer Membrane Protein Import Channels

### 3.1. Tom40 Channel

Tom40 is the central and channel-forming subunit of the TOM complex [3,21]. A wide range of important studies on protein import into mitochondria have shown that Tom40 is not only the entry gate for most mitochondrial precursor proteins delivered from the cytosol to various sub-mitochondrial locations [39,40,41] since it also works as a decisive general selectivity filter in the uptake of newly synthesized mitochondrial proteins [42,43]. Moreover, it was also proposed that the TOM complex can act as an insertase mediating lateral release of the membrane imported proteins into the outer membrane [44].

As a protein with a β-barrel structure, Tom40 [45] contains 19 antiparallel β-strands that enable channel formation [10]. The copy number of Tom40 proteins in yeasts cells was estimated at ~45,000 copies per cell [46]. The structure of the TOM complex and its electrophysiological properties has been studied for the last three decades. In the previous study [47,48], two kinds of the TOM complex were observed: three-channel complex representing trimeric form of Tom40 and two-channel complex representing dimeric form of Tom40. It was showed that the two-channel complex consisted of two Tom40 and small Tom subunits but did not contain Tom22. It was concluded that the two-channel complex dimmer functioned as a premature complex [49] whereas the mature TOM complex dynamically exchanges with the two channel one providing an assembly platform for the integration of new subunits. Recent years have brought the intensive development of microscopy techniques that have facilitated the study of specifically labelled active membrane protein complexes in the native membrane. The structure of the studied complexes is based on electron cryo-tomography (cryo-ET) presenting the surface and internal conformation at nanometre resolution [50]. The higher resolution of cryo-EM microscopy revealed a stable dimeric form of the TOM complex for *Neurospora crassa* at a resolution of 0.68 nm [51], with resolutions of 0.38 nm for *S. cerevisiae* [52] and 0.34 nm for humans [53] and a resolution of 0.41 nm for its tetrameric form [50,54]. According to the obtained TOM complex model, two Tom40 subunits form two pores for protein translocation across the membrane. Each Tom40 is connected by two Tom22 receptors and one phosphatidylcholine (PC) molecule. The latter is the most abundant phospholipid in the mitochondrial outer membranes and functions as a player in protein import into mitochondria [55]. Each Tom40 is surrounded by small Tom subunits (i.e., Tom5, Tom6 and Tom7), with Tom5 being situated at the distal end of the Tom40 dimer and Tom6 and Tom7 being located on opposite sides across the channel pore. A study on the human TOM complex [53] revealed a negative interior of the pore and positive regions at its periphery. Notably, the dimeric form of the TOM complex may associate to form a tetramer. The molecular weight of the human TOM complex was estimated at 150 kD, which is comparable to the size of the TOM complexes of *S. cerevisiae* [52,54] and *N. crassa* [51,56]. This is congruent with a previous study on the monomeric forms of the *N. crassa* [47] and *S. cerevisiae* TOM complexes [40], which were obtained via the circular dichroism method. These TOM complexes were estimated at a resolution of 0.68 nm and showed two pores in the dimer, each with a shortest diameter of 1.1 nm and longest diameter of 3.2 nm.

The structure of the Tom40 pore has been reported by various groups [52,54] and resulted in similar models with some differences. In contrast to the Tom40 structure of yeast, the human Tom40 structure does not contain an α-helical segment preceding the internal helix at the N terminus and also lacks an additional C-terminal α-helix at the end of 19th β-strand. Studies of the purified Tom40 of *Candida glabrata* [57] revealed that the conserved 51 amino acids at the N terminus of the Tom40 sequence, the 15 amino acids at its C terminus and the interaction with a pre-sequence of the precursor protein were not crucial for channel formation. These results demonstrated that all studied Tom40 constructs formed channels of different conductance. Each of the expressed proteins interacted with a pre-sequence peptide in the concentration dependent manner although the interaction with the conserved N-terminal Tom40 domain was not required. This study showed that Tom40 dimers were functional and their distinct conformations were regulated by binding of the substrate. For yeast and human Tom40, the α-helical domain that traverses the channel pore and extends into IMS plays a crucial function in import across the Tom40 channel. It was noted that residues T82, N79, P80, H87 and Q97 in the internal portion of Tom40 form hydrogen bonds with R195, T200, K253, T264 and Q353 in β-strands 7, 8, 11, 12 and 19 of Tom40, respectively. Despite it being partially unfolded, the presence of the internal helix limits the diameter of Tom40 enough for the translocation of the precursor protein.

As previously mentioned, the activity of the Tom40 channel could be stimulated by PC, which contains the positively charged choline group extending into the IMS. This was suggested to have a possible function in the translocation process [55]. Electrophysiological studies were performed on purified TOM complex and purified Tom40. The Tom40 channel shows a preference for cations: *P_K_^+^*/*P_Cl_^−^* = 8:1 with a reversal potential of 40 mV (planar lipid bilayer technique performed on *S. cerevisiae* Tom40, according to [40]) (Table 1). In an experiment performed by [58], *S. cerevisiae* and *N. crassa* Tom40 were expressed in *Escherichia coli* cells and renatured. The electrophysiology measurements in planar lipid bilayers at a symmetrical concentration of 250 mM KCl were indistinguishable at 370 and 390 pS, respectively.

Tom40 is the voltage dependent channel. In the absence of membrane potential, the reconstituted channel is completely open but closes symmetrically at positive or negative potentials. The studies on the reconstituted *S. cerevisiae* Tom40 showed that in the presence of low membrane potential the pre-sequence peptide selectively bound to Tom40 and thereby altered the gating of the channel but was rapidly translocated through the channel when driven by higher voltage [40]. For purified and reconstituted *S. cerevisiae* TOM complex, the maximal conductance was estimated at 740 ± 18 pS, whereas the same estimate in the outer membrane vesicles was 760 ± 12 pS. In general, the maximal conductance of the *S. cerevisiae* or *N. crassa* TOM complexes were 2-fold higher than the total conductance of renatured Tom40 alone. The maximal conductance channel was 370 ± 8 pS for *S. cerevisiae* Tom40 and 390 ± 10 pS for *N. crassa* Tom40 at 250 mM KCl. Thus, the channel properties of purified Tom40 are lower than those of the TOM complex [58]. Interestingly, in the case of the *A. castellanii* and *D. discoideum* TOM complexes, channel conductance activities were estimated at 2.5 and 2.3 nS, respectively, in 1M KCl [59]. After extrapolation (625 pS and 575 nS respectively in 250 mM KCl), this result is in agreement with the data obtained for *S. cerevisiae* and *N. crassa*. The data for *N. crassa* [60] correspond well to the conductance states for *C. glabrata*, where different truncated forms of purified Tom40 were studied in planar lipid bilayers (~125–500 pS in 250 mM KCl, described by [57] (Table 1). The reconstituted version of *C.glabrata* Tom40 forms provided a better understanding of the channel flickering phenomena observed during a study of *S. cerevisiae*, *N. crassa*, *A. castellanii* and *D. discoideum* TOM complex [58,59]. This data indicated flickering which is typical for this kind of the outer membrane channels.

The channel conductance was measured using the specific signal peptides known to be recognised by the TOM complex receptors [61]. The presence of signal peptides resulted in the channel flickering, i.e., rapid transitions from the fully open state to substates of lower conductance resulting from the channel blockade. This is a consequence of signal peptides interaction with TOM complex subunits. In a study by [58], it was shown that for the purified Tom40 of *S. cerevisiae*, the signal peptide of CoxIV had to be used in a much higher concentration than in the case of the TOM complex to obtain a 50% reduction in open channel probability (3 µM vs. 50–80 nM). This implies that the sensitivity of the Tom40 channel pore is enhanced by the presence of the TOM complex subunits. The channel properties were also measured in the presence of the other recognised signal peptide. The pre-sequence peptide of the F1-ATP synthase subunit β also induced the blockage of Tom40 conductance in a concentration-dependent manner, which indicates a direct interaction of this peptide with the Tom40 channel pore. Similarly, in a study of the *A. castellanii* and *D. discoideum* TOM complexes, a signal peptide of subunit 9 of ATP synthase also blocked the purified TOM complex in a concentration-dependent manner [59]. Additionally, the results obtained for the TOM complex revealed that the binding sites for the incoming precursor proteins had a presequence recognised by the TOM complex subunits, which was not present in purified Tom40. Notably, data on the individual TOM complex receptors revealed relatively low affinities (high KD) for the pre-sequences [40,62], while high-affinity binding of the pre-sequences was observed for the TOM complex [61]. Thus, data on the TOM complex and purified Tom40 alone uncovered the roles of the TOM complex subunits in the regulation of Tom40 channel activity.

It should be mentioned that the means of protein preparation for the studies on Tom40 and other described in this review protein channels, as well as the applied methods as planar lipid bilayer membranes and patch clump could greatly influence the determined channel properties. Thus, combining electrophysiological data with structural data provide reasonable background for putative modulation. However, this always would need verification under cellular conditions.

### 3.2. Tob55/Sam50 Channel

Tob55/Sam50 protein (also known as Omp85) is a member of the Omp85 family of β-barrel-channels typical of the outer membranes of Gram-negative bacteria, mitochondria and chloroplasts [25,45,77]. The copy number of Tob55/Sam50 proteins in yeasts is estimated at ~1500 copies per cell [3,46]. The Tob55/Sam50 channel is composed of a 16-stranded transmembrane β-barrel with a single polypeptide-transport-associated (POTRA) domain extending into the IMS. The POTRA domain is an N-terminal domain consisting of ~100 amino acids involved in the recognition and transfer of β-barrel forming precursors from the IMS via small Tim9 and Tim10 to the TOB/SAM complex [8,9,10,11].

The TOB/SAM complex was also suggested to be involved in apoptosis [72]. An important study on yeast and human cells conducted by [72] revealed that Tob55/Sam50 was required for the transport of granzymes (a and b) and caspase-3 to cross the outer membrane. Granzymes are serine proteases that trigger cell death in a caspase-dependent and independent manner [78,79]. It was also shown that Tob55/Sam50 was sufficient for active caspase-3 to enter the mitochondria to induce cell death. Moreover, the depletion of Tob55/Sam50 caused cells to become more resistant to cell death. Thus, Tob55/Sam50 is engaged in a wider range of imported substrates than previously thought (Table 1).

The TOB/SAM complex remains in dynamic interactions with other complexes, including TOM [80,81], MICOS (e.g., [82]) and ERMES (shared Mdm10 subunit) [83]. The TOM complex binding site is based on the interaction between Tom22 and Tob37 [29,81] and on the interaction between Tom5 and Tom22. It was found that a small fraction of Tom5 being associated with a TOB/SAM complex promotes the folding of Tom40 during its import to the outer membrane [80,84]. The interaction of the TOB/SAM complex with the MICOS complex is based on Mic60/mitofilin [36,85,86] and results in the regulation of cristae morphology, mitochondrial shape and respiratory chain complexes’ assembly [85,86,87]. It has been shown that the TOB/SAM complex subunits exist in a large protein complex with Mic60/mitofilin, other Mic proteins (i.e., Mic10, Mic12, Mic19, Mic23, Mic25 and Mic 27) and CHCHD3 to create the so-called large mitochondrial IMS bridging (MIB) complex [85,86,88]. The MIB complex also likely contains the homolog metaxin-3 the DnaJC11 protein [89]. The depletion of Tob55/Sam50 in human cells resulted in the loss of the crista junction, which highlights its crucial function in crista formation [89]. The interaction of TOB/SAM with ERMES is mediated by Mdm10, which is considered a subunit of both of these complexes [82,90,91].

The Tob55/Sam50 channel has an inner diameter of approximately 7–8 nm (outer diameter: ~15 nm, central cavity: ~4–5 nm) [3,11]. A recent study on the *S. cerevisiae* TOB/SAM complex performed by electron microscopy at a resolution of 0.28–0.32 nm resulted in the description of two different forms of the TOB/SAM complex [92]. The TOB/SAM complex forms a dimer based on two different Tob55/Sam50 isoforms: Tob55/Sam50a and Tob55/Sam50b. Tob55/Sam50a is responsible for channel formation, while isoform b is responsible for releasing incoming precursor proteins from the channel [92]. The *S. cerevisiae* Tob55/Sam50 channel was studied in planar lipid bilayers, which revealed that it displayed a conductance of 640 pS at 250 mM KCl [71]. This result is in agreement with data obtained by [8] where Tob55/Sam50 showed a channel activity level of 3.75 nS in 1M KCl what after extrapolation correspond to 925 pS in 250 mM KCl Tob55/Sam50 has a preference for cations (*P_K_^+^*/*P_Cl_^−^* = 4:1) and a reversal potential of 30 mV [8,71]. At voltages greater than ±70 mV, the channel formed by Tob55/Sam50 is partially closed, which clearly distinguishes this channel from the VDAC channel measured by the same technique as planar lipid bilayers [93]. It has also been shown that the channel is mediated by a precursor protein β-signal that causes the displacement of the endogenous carboxy-terminal β-signal of the Tob55/Sam50 channel [94]. β-signal is localised at the C-terminal part of β-barrel precursor proteins and is highly conservative among bacteria, mitochondria and chloroplasts [71,94]. β-signal was shown to interact with Tob55/Sam50 channel and bind precisely at its β-strand 1. In doing so, it replaces β-strand 16 (endogenous Tob55/Sam50 β-signal). This opens the channel’s lateral gate between β-strand 1 and β-strand 16 [94]. From this point, the remaining precursor proteins are inserted to gradually gain β structures. At the end of this process, matured protein is released into the outer membrane of the mitochondria [29,94,95] (Table 1).

### 3.3. Mdm10 Channel

Mdm10 has no sequence homologs in bacteria and is not present in higher eukaryotes [10]. In *S. cerevisiae*, its molecular weight is 56 kDa and the copy number is estimated at ~500 copies per cell [46]. Mdm10 was found together with Mdm12, Mdm34 and Mmm1 as a part of a complex responsible for the maintenance of mitochondrial morphology and distribution [96]. Mdm10 is involved in lipid biosynthesis and the tethering of the endoplasmic reticulum (ER) with mitochondria [83]. The resulting connection between these organelles is called the ERMES complex [97]. In addition, being a part of the ERMES complex, Mdm10 protein was reported to bind to the TOB/SAM complex and influence β-barrel protein import into the mitochondrial outer membrane [83,90,91]. After being trans-ported through the TOB/SAM complex into the TOM, Mdm10 is involved in the late assembly of the TOM complex by interacting with the TOB/SAM complex in the process of proper Tom40 protein assembly [83,90,98]. This step involves the association of Tom40 with Tom22 and small Tom proteins. This interaction between Mdm10 and TOB/SAM is regulated by Tom7 protein, which stimulates the release of Mdm10 from the TOB/SAM complex [3,28,98,99].

After incorporation into liposomes and planar lipid bilayers, the expressed and purified *S. cerevisiae* Mdm10 exhibits channel activity with a main conductance of 480 pS at 250 mM KCl and has a preference for cation selectivity (*P_K_^+^*/*P_Cl_^−^* = 2.8:1; reversal potential: 21.5 mV). Moreover, three independently gated pores have been reported [14]. Additionally, the presence of Tom22 precursor increases Mdm10 conductance to 550 pS at 250 mM KCl while also increasing the number of independently gated pores from three to four. This indicates the possible function of Mdm10 in the import of Tom22 protein [4,14].

### 3.4. Mim1 Channel

Mim1, also known as Tom13, plays a role in the import of α-helical outer membrane proteins such as Tom70 and Tom20 [12]. This protein forms a channel composed of the N-terminal domain exposed to the cytosol, central putative transmembrane segment (TMS) and C-terminal domain exposed to the IMS [12,13]. Mim1 is involved in the import of small Tom proteins and indirectly influences proper Tom40 membrane insertion [100,101] because the N-terminal domain of Mim1 and Tob37/Sam37 combined regulate the release of Tom40 protein from TOB/SAM [13,100]. Mim1 is also crucial for the proper insertion of Tom70 and Tom20, which, in the case of the latter, requires the homo-oligomerisation of Mim1 proteins via its transmembrane domain (TMS) [12]. It has also been shown that the precursors of multi-spanning α-helical proteins directly interact with Mim1, which cooperates with the Tom70 receptor for protein insertion into the outer mitochondrial membrane [101]. Moreover, Mim1 regulates the binding and insertion of mitochondrial fusion and the transport protein Ugo1 [102] while promoting the import of UBX domain-containing protein 2 (Ubx2), which is the TOM complex-associated protein suspected to remove mitochondrial precursor proteins that become stuck in the import channel [103].

The *S. cerevisiae* Mim1, expressed in *E. coli* and then purified and renatured, was studied in planar lipid bilayer. A channel conductance of 580 pS was calculated at 250 mM KCl and a closing tendency due to high positive and negative voltage was observed. This study showed cation selectivity (*P_K_^+^*/*P_Cl_^−^* = 23.5:1) and a reversal potential of 53 mV [4]. The channel activity was modulated by specific anti-Mim1 antibodies. The same channel properties were obtained for *S. cerevisiae* Mim1 expressed in wheat germ lysate [4]. Modulation of the Mim1 channel was also observed after adding purified Mim2, which did not reveal channel activity. The channel properties of Mim1 determined in the presence of Mim2 included a reduced maximal current, decreased reversal potential of 48 mV and reduced cation selectivity *P_K_^+^*/*P_Cl_^−^* of 11:1. Thus, Mim1 is a channel of cation preference for the positively charged precursor proteins modulated by the Mim2 protein [4].

### 3.5. Mitochondrial Apoptosis-Induced Channel (MAC)

MAC is the outer membrane large channel known as an early marker of the onset of apoptosis. It participates in release of proteins normally constrained within the intermembrane space, such as cytochrome c, second mitochondria-derived activator of caspases (Smac)/Direct inhibitor of apoptosis-binding protein with low pI (DIABLO) or apoptosis induced factor (AIF) [75,104]. MAC is formed by Bax and/or Bak proteins and at least one of the proteins must be present. Bax protein is cytosolic whereas Bak is an integral protein of the outer membrane and they both stay in an inactive form until MAC formation [75,104]. The activity of MAC is significantly different from the TOM and TOB/SAM complexes. Moreover, the former are constitutive channels of the outer membrane whereas MAC activity co-occur only with apoptosis [75,76,104]. MAC activity is regulated by other Bcl-2 family proteins. Bcl-2 and Bcl-xL inhibit apoptosis by separating Bax and Bak. Moreover, MAC can be block by dibucaine, trifluoperazine and propranolol in a dose-dependent manner) as well as 3.6 dibromocarbazole piperazine derivatives of 2-propanol (that blocked cytochrome c release induced in isolated mitochondria by tBid).

By application of patch-clump, the reconstituted MAC conductance was estimated as heterogenous; i.e between 1.5–5 nS (1500–5000 pS in 150 mM KCl). MAC frequently fluctuates between the fully open state and fully closed state, with a maximal single transition size of 2000 pS in 150 mM KCl and at least three substates [75]. MAC is a voltage-independent channel and is slightly cation-selective. As it has been shown for mammalian apoptotic FL5.12 cells after interleukin 3 (IL-3) withdrawal, *P_K_^+^*/*P_Cl_^−^* = 3:1. The cation selectivity is consistent with its putative role in releasing cationic proteins such as cytochrome *c* [75]. At high conductance state MAC is permeable dextran of molecular weight in the range of 10 to 17 kDa, but not in the range of 45 to 71 kDa. Based on the polymer exclusion method, MAC pore diameters were estimated in the range of 2.9 to 7.6 nm [75,76].

## 4. Non-Proteinaceous Molecules Transported by Protein Import Channels

### 4.1. Transport of RNA

RNA import into mitochondria has been shown for many different groups of eukaryotic organisms, including plants, mammals, yeast *S. cerevisiae* and protozoans. Notably, it is assumed that this phenomenon is universal for all eukaryotes. It has also been shown that mitochondria can import diverse types of RNA molecules, which suggests the existence of an extrinsic RNA importome [105]. The latter includes transfer RNAs (tRNAs), ribosomal RNAs (rRNAs), microRNAs (miRNAs) and long non-coding RNAs (lncRNAs) (for a review on this topic, see [105,106,107]). RNA import and the imported RNA molecules’ contribution to gene expression are considered essential for mitochondrial function (e.g., [108]). The majority of the studied imported RNA molecules are tRNAs (e.g., [109,110]) since their import can be clearly explained as a compensation for tRNA encoding genes not being present in the mitochondrial genome (mtDNA). This applies to plants lacking a few tRNA encoding genes in mtDNA (e.g., [111]) as well as protozoans lacking a distinct part or complete set of tRNA coding genes in mtDNA (e.g., *Tetrahymena thermophila*, *Trypanosoma brucei* and *Leishmania tarentolae* [112,113,114]). Moreover, tRNA import from the cytosol can be observed in the presence of all necessary tRNA genes in mtDNA and is regarded as an important mechanism in stress response [105,115].

Available data indicates that routes for RNA import into the mitochondria may overlap with mitochondrial protein import channels since some components of the TOM and TIM complexes may contribute to translocation across mitochondrial membranes. In the case of the mitochondrial outer membrane, translocation is purportedly mediated by Tom40; however, an important role is also assigned to relevant VDAC paralogs. In plant mitochondria, Tom40 and Tom20 likely participate in the fixation of RNA molecules at the surface of mitochondria [63,64], while Tom40 is also suggested to partially contribute to translocation in yeast and mammalian mitochondria [105,109,110,116,117]. In the case of African trypanosomes (represented by *T. brucei*), the Tom40 orthologue ATOM40 (atypical TOM, e.g., [118]), as well as Tom22 orthologues ATOM14 and two additional ATOM subunits (ATOM11 and 12), have been shown to perform RNA molecule translocation across the mitochondrial outer membrane [63,119]. While the proteins involved in RNA molecule translocation across the inner mitochondrial membrane remain largely unknown, data available for yeast and *T. brucei* point to the important contribution of TIM complexes. In the case of yeast mitochondria, TIM23 is suggested as the translocation pathway due to the putative involvement of Tim44 (subunits of the TiM23 complex) [105,116,120]. For *T. brucei*, the Tim22 orthologue TbTim17 (subunit of the TIM22 complex) has been shown to participate in translocation [120]. For translocation, the latter requires some of the other subunits of the non-canonical, singular TIM complex [121] (i.e., TbTim42, Tb-Tim62 and putative acyl-CoA dehydrogenase homologue (ACAD) [122].

The mechanisms involved in the import of RNA molecules into mitochondria are still not completely understood and seem to differ between organisms with different phylogenetic lineages and RNA molecule types. Although the transport of molecules across mitochondrial membranes is ATP-dependent and requires the presence of the inner membrane potential and relevant translocating proteins, differences exist in terms of selective import signals and the interacting proteins participating in redirection to the mitochondrial surface (e.g., [105,107,110]). In the case of imported tRNA molecules, the participating proteins may involve aminoacyl-tRNA synthetases in plants and yeast (e.g., [115,123], respectively), cytosolic translation elongation factor eEF1α in *T. brucei* [124] and glycolytic enzyme enolase in yeast [125]. It is also suggested that mitochondria-targeted proteins may be common elements participating in the redirection of different RNA molecules to mitochondria and their translocation across membranes (e.g., [105]). However, it is also suggested that RNA molecules may be translocated by Tom40 without protein assistance [63]. There is also the possible contribution of mitochondrial IMS proteins, which could function as intramitochondrial RNA import factors, as proposed for polynucleotide phosphorylase (PNPase) in human mitochondria (e.g., [105]).

### 4.2. Transport of Metabolites

It is well known that the universal pathway for metabolite transport across the outer mitochondrial membrane is formed by the VDAC, which may be formed by different VDAC paralogs (e.g., [126,127,128,129,130]). The molecular mass cut-off for the transported molecules is assumed to be about 4 kDa although the limit could be decreased when VDAC switches to lower conducting substates featuring less anion selectivity [131]. Nevertheless, VDAC is considered to be the major transport pathway for compounds as diverse as inorganic ions (e.g., K^+^, Na^+^ and Cl^−^), metabolites of different size and charge (e.g., big anions as ATP, AMP and glutamate and small anions as superoxide anion as well as big cations as NADH and acetylocholine) and large macromolecules such as tRNA (e.g., [128,132,133]). However, for yeast *S. cerevisiae* mitochondria under the condition of the dominant VDAC paralog(yVDAC1) limited permeability or encoding gene deletion, metabolite transport across the mitochondrial outer membrane may be supported by the TOM complex [68,69,134]. This assumption was initially based on observation that in the absence of yVDAC1, NADH, ADP and CATR (carboxyatractylate) displayed limited access into the mitochondrial intermembrane space and the observed limitations depended on charge and size of these molecules (the highest for CATR being big anion of molecular weight higher than ADP and the lowest for NADH being big cation of molecular weight slightly smaller than CATR). Moreover, the limitations were weakened by the presence of Mg^2+^ that together implies involvement of cation selective channel of lower conductance that yVDAC1 [134]. This in turn correlates with the electrophysiological characteristic of the TOM complex channel (see Section 3.1 for details).

Later, it has been proven that blockage of the TOM complex by an imported protein decreases external reduced nicotinamide adenine dinucleoide (NADH) access to the inner membrane, which limits imported protein translocation by the TOM complex and increases competition in the absence of functional yVDAC1. Moreover, it has been observed that blocking the TOM complex with an imported protein decreased superoxide anion (O_2_^•−^) release from mitochondria, particularly in the absence of functional yVDAC1 [135]. Notably, yeast mitochondria contain two VDAC paralogs (i.e., yVDAC1 and yVDAC2 [136,137]), both of which can form channels of comparable electrophysiological characteristics [130], but differ in their expression levels and (likely) substrate selectivity [138]. These postulated differences coincide with a yVDAC2 contribution to the global permeability of the mitochondrial outer membrane that has remained undetected to date [4,139]. While yVDAC2 encoding gene deletion also results in competition between imported proteins and external NADH for access to the yeast mitochondria inner membrane, this competition is distinctly less pronounced than in the absence of functional yVDAC1 [68]. This observation contributes to solving the unresolved issue of yVDAC2 function.

One of the regulatory aspects concerning TOM complex involvement in metabolite transport is the expression level of its subunits. It has been reported that the expression level increases in the absence of yVDAC1 or yVDAC2, with the latter increase being less pronounced [68,73]. The upregulation of TOM complex subunits may result from increased transcription as well as the stability and/or translation of proper mRNAs [140]. Interestingly, this upregulation appears to be mediated by the cytosol reduction-oxidation (redox) state change in the absence of yVDAC1 or yVDAC2 and during yeast cell growth [73,74]. Moreover, redox state-dependent upregulation was also observed for Tob55/Sam50, which implicates TOB/SAM complex involvement in metabolite transport [73,74,139]; however, functional data to support this involvement remain missing.

### 4.3. Ayr1 and OMC7 and OMC8: Mitochondrial Outer Membrane Channels for Unknown Molecules

In the mitochondrial outer membrane, three channels of unknown function have been identified. These include a cation-selective channel formed by Ayr1 and two channels of anion selectivity formed by OMC7 and OMC8 [4]. *S. cerevisiae* Ayr1 (1-acyldihydroxyacetone-phosphate reductase) is a 33kD protein containing a conserved nucleotide-binding motif (TGX3GXG) that is localised in the ER [141,142]. It was suggested that this could be part of the contact sites between the ER and mitochondria [143]. This protein has also been suggested to participate in cell wall biogenesis and lipid metabolism [144]; however, its true function remains unexplained.

Ayr1 reconstituted in liposomes in planar lipid bilayer membrane revealed a main conductance of 1.47 nS in 1 M KCl (367 pS at 250 m M KCl after extrapolation) in the fully open state. Additionally, three partially open states with a lower conductance of ≅490 pS at 250 mM KCl and a reversal potential of 30 mV were observed. It has also been shown that the channel, as a monomer, displays cation selectivity (*P_K_^+^*/*P_Cl_^−^* = 4.5:1) that can be modulated by NADPH, which increases the reversal potential to 43 mV and cation selectivity ratio *P_K_^+^*/*P_Cl_^−^* to 10:1 [4]. The observed cation selectivity of the Ayr1 channel might suggests its contribution to protein import. The presented parameters of the Ayr1 could be consistent with a localization of Ayr1 in mitochondrial outer membrane and ER fractions, as being arranged in contact sites between ER and mitochondria.

In contrast, OMC7 and OMC8 were described as anion-selective channels of similar conductance (550 and 570 pS, respectively, at 250 mM KCl) (Table 1). In light of its anion selectivity, these channels could not be suggested as being engaged in the interaction with positively charged precursor protein; however, its contribution to nucleic acid and metabolite transport within and outside of the mitochondria are worthy of further study.

## 5. Conclusions and Perspectives

The mitochondrial outer membrane selectively communicates with the cytosol environment via protein channels for a wide variety of molecules that must be delivered into the mitochondria and/or released from the mitochondria. Moreover, these channels display high flexibility in their specificity, as illustrated by their ability to transport different types of molecules. Undoubtedly, these results highlight many implications for the channel modulation that might affect mitochondrial protein turnover as well as transport of different small molecules including crucial metabolites as ATP, ADP, substrates of the respiratory chain and superoxide anion, as well as drugs as granzymes. This in turn might be beneficial for treatment of mitochondria-linked diseases mediated by the outer membrane permeability impairment as cancer [75] as well as by incorrect intramitochondrial processes as neurodegenerative disorders including Parkinson and Alzheimer diseases.

## Figures and Tables

**Figure 1 molecules-26-04087-f001:**
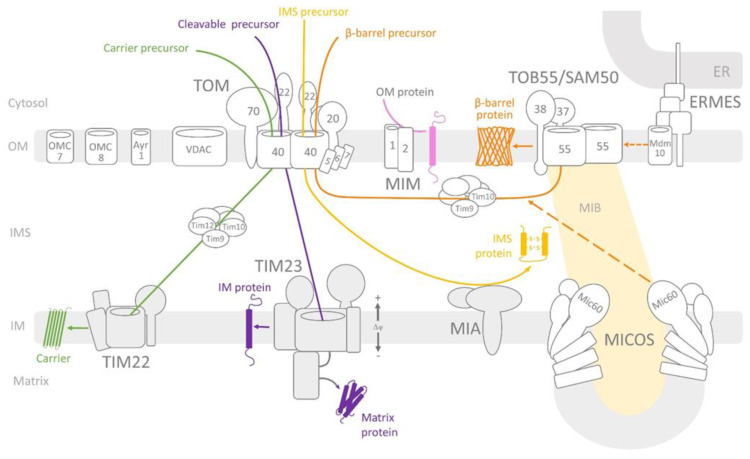
Overview of protein import into mitochondria. The TOM complex recognises most of the mitochondrial pre-proteins via its receptors (i.e., Tom20, Tom22 and Tom70) and imports across the outer membrane through the Tom40 channel (dimer). Small TOM subunits (i.e., Tom5, Tom6 and Tom7) stabilise the TOM complex. β-barrel precursors (i.e., Tom40, Tob55/Sam50, VDAC and Mdm10) are inserted into the outer membrane by the TOB/SAM complex, which consists of the protein channel Tob55/Sam50 (dimer) and receptors Tob38 and Tob37. The mitochondrial contact site and cristae organising system (MICOS) complex supports the insertion of the β-barrel proteins into the OM. The dynamic structure of TOB/SAM and MICOS is called mitochondrial intermembrane space bridging (MIB). α-helical precursors are inserted into the OM by the MIM complex (Mim1 and Mim2). The ERMES complex links the mitochondria and endoplasmic reticulum (ER) and supports the topogenesis of the mitochondrial outer membrane β-barrel proteins/sorting and assembly machinery (TOB/SAM) complex. The precursor targeted into the intermembrane space engages the MIA complex. TIM23 is responsible for import across the IM or into the mitochondrial matrix. TIM22 inserts carrier proteins into the IM. Voltage-dependent anion-selective channel (VDAC) is responsible for metabolite transport. Ayr1, OMC7 and OMC8 are the channels of unknown substrates. OM—outer membrane; IM—inner membrane; IMS—intermembrane space; ER—endoplasmic reticulum.

**Table 1 molecules-26-04087-t001:** Characteristics of the mitochondrial outer membrane channels based on reconstitution studies *.

Protein Name	Conductance **	Reversal Potential (mV) ***	*P_K_^+^* */P_Cl_^−^*	Studied Organism	Constitutively Open	Molecules Transported across Channel	Channel Activity Modulators	References
Tom40	370 ± 8 pS ^a^	40 ^a^	8 ^a^	*S. cerevisiae* ^a^	Yes	Mitochondria proteins ^c^ RNA ^d^ Pink 1 ^e^ Aβ ^f^	PC ^i^ α-Syn ^j^	[58] ^a^ [59] ^b^ [6] ^c^ [63] ^d^ [64] ^e^ [65] ^f^ [66,67] ^g^ [68] ^h^ [69] ^h^ [58] ^i^ [70] ^j^
	390 ± 10 pS ^a^			*N. crassa* ^a^				
	TOM complex: 740 ± 18 pS (purified complex) ^a^ 760 ± 12 pS (complex in OMVs) ^a^			*S. cerevisiae* ^a^	Yes	TOM complex: RNA ^g^ Metabolites ^h^	signal peptides of: CoxIV ^a^ preSu9 ^a,b^	
	625 npS (purified complex) ^b^			*A. castellanii* ^b^				
	575 pS (purified complex) ^b^			*D. discoideum* ^b^				
Tob55/ Sam50	640 pS ^k^	30 ^k^	4 ^k^	*S. cerevisiae* ^k^	Yes	β barrel proteins ^l^ Granzymes (a and b) ^m^ Caspase-3 ^m^ Suggested: metabolites ^n^	Unknown	[71] ^k^ [30] ^l^ [72] ^m^ [73,74] ^n^
Mdm10	480 pS ^o^	21.5 ^o^	2.8 ^o^	*S. cerevisiae* ^o^	Yes	Unknown	Tom22 ^o^	[14] ^o^
Mim1	580 pS ^p^	53 ^p^	23.5 ^p^	*S. cerevisiae* ^p^	N/D	α helical outer membrane proteins ^r^	Mim2 ^p^	[4] ^p^ [12] ^r^
MAC	1500–5000 pS ^s^	voltage-independent ^s^	3 ^t^	Mammalian ^s^	No	apoptotic cofactors e.g.,: Cyt c ^s^ Smac/DIABLO ^s^ AIF ^s^	Bcl-2 ^s^ Bcl-xL ^s^ Dibucaine ^s^ Trifluoperazine ^s^ Propranolol (and its derivatives) ^s^	[75] ^s^ [76] ^t^
OMC7	570 pS ^p^	−12.5 ^p^	0.55 ^p^	*S. cerevisiae* ^p^	N/D	Suggested: RNA and/or metabolites	Unknown	[4] ^p^
OMC8	550 pS ^p^	−15.5 ^p^	0.48 ^p^	*S. cerevisiae* ^p^	N/D	Suggested: RNA and/or metabolites	Unknown	[4] ^p^
Ayr1	1470 pS ^p^	30 ^p^	4.5 ^p^	*S. cerevisiae* ^p^	N/D	Suggested: proteins	NADPH ^r^	[4] ^p^

* Channels are listed in the order in which they appear in the text. Abbreviations: PC—Phosphatidylcholine; α-Syn—α-synuclein; Aβ—amyloid β peptide; Pink 1—PTEN-induced kinase 1, cyt c—cytochrome c, AIF—Apoptosis Inducing Factor. References are listed using letter designations (a–t), with each letter also being assigned to the data contained in a given publication. ** Study performed in 250 mM KCl (by planar lipid bilayer membranes), with exception of MAC channel: 150 mM KCl (study performed by patch clump). *** Study performed in 250 mM/20 mM KCl (cis/trans) gradient.

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
