# Peer review of "The Diversity of the Mitochondrial Outer Membrane Protein Import Channels: Emerging Targets for Modulation"

_molecules, 2021, doi:10.3390/molecules26134087_

Round 1
Reviewer 1 Report
In this manuscript, Mazur et al review the currently known knowledge about the functions of mitochondrial outer membrane proteins involved in proteins and metabolites (i.e molecules, ions and nucleic acids) import. In this review, the authors have described in detail the role of outer mitochondrial membrane channels, focusing on their structure and regulation (and variety of transported molecules). Furthermore, the authors well summarized the protein import machinery as illustrated in figure 1. The manuscript is overall clear written and should be considered for publication after some typewriting revision (i.e. lanes 77, 113, 301, 302, 362, 432).
Author Response
We would like to thank the Reviewer for the positive comments concerning our manuscript. Each of the pointed typewriting errors were corrected.
Reviewer 2 Report
Mitochondrial channels play numerous and most probably yet undefined functions, decisive for cell’s normal behavior. This is a well written and focussed review that summarizes some studies describing the most outstanding structural and functional features of the large channels and protein import related complexes found in the mitochondrial outer membrane.
This is a good, clearly presented and up to date revision that deserves to be published, as it has the potential to interest a wide audience, including scientists not clearly involved in the field. However, a number of changes should be considered to improve the quality and accuracy of the manuscript:
1) According to the title, membrane permeability modulation is one of the prominent aspects of this manuscript. However, this issue is merely scattered, and thus blurred throughout the text. More emphasis or a devoted section to this growing up theme might be considered.
2) Although the information is exact, there is a weakness that should be underlined. The authors do not cover any aspects, electrophysiological nor biochemical, of the Mitochondrial Apoptosis-induced Channel (MAC), a large protein transport channel also identified in the outer membrane.
3) The channel of the TOM complex is reported as the dimeric or tretrameric forms of Tom40. However, the trimeric form, its dynamics and function are ignored (e.g. Shiota, T. et al. Molecular architecture of the active mitochondrial protein gate. 2015. Science 349: 1544–1548. doi:10.1126/science.aac6428).
4) In line with the above, evidence for structurally distinct Tom40 conformations is mentioned (e.g. ref. 54), but their dynamic implications in substrate recognition and in transport function are largely omitted.
5) In general, bibliographic references are timely and well incorporated into the review. However, citations concerning functional aspects (namely the electrophysiology) are scarce in relation to those referring to structural aspects.
6) The electrophysiology of Tom40 is confusing. E.g. the way to report the size of the channel in different organisms is tangled by the different salt concentrations used in electrophysiological approaches and misleads the non-expert readers. Too many distracting details sidetrack the reader from the main message, i.e. a similar conductance size.
Table 1 does not settle much the issue of significance or relevance of this type of studies, as conductance, reversal potential or permeability ratios are not discussed nor interpreted in light of their possible functional significance. By the way, at this level, reversal potential and ion permeability ratios, basically provide the same information. Whereas other relevant channel properties (e.g. voltage dependence, signal peptide sensitivity, number of substates etc.) are omitted. Also on this table, differences between transported molecules and modulators of the channel activity is not too precise. For instance, Aβ is under the modulator column, whereas ref. 67 points to it as transported molecule. Finally, the heading of this table is striking. I do not recall any mitochondrial outer membrane channels experiments performed “in vivo”.
7) Electrophysiological studies should bring for more than just a list of physical characteristics of the channels. Functional implications of these characteristics are lacking and must be outlined (e.g. what is the relationship between the size or the fully open and the half open states of the channel, and the size of a protein in transit through the channel? Are the channels always open? What triggers the gating of the channels? Besides, all these are important aspects related to membrane permeability regulation.
In line with some inaccuracies found in the electrophysiology, the statement about flickering (lines 189-191), as it is (i.e. without further interpretation) is irrelevant.
Statement on lines 192-193 is also misleading: Signal peptides do not measure the regulation of channel conductance (rather than conductivity). Signal peptides trigger the channel activity by increasing its flickering, i.e. rapid transitions from the fully open to the substate and close state. Even more, ref. 58 does not refer to this aspect but rather to the recognition of preproteins by the isolated TOM complex.
No mention is given to whether the channel remains or not closed in the absence of signal peptides/preproteins and if signal peptides/preproteins are the ones triggering the opening of the channel.
8) It is well known than the reconstituted system (i.e. isolated protein, purified complex, proteoliposomes, or native mitochondrial membranes) as well as the electrophysiological technique applied (i.e. planar bilayer vs patch-clamp) greatly influence the outcome of the channel properties obtained. Thus, dramatic differences in voltage dependence, ion selectivity, reversal potential, flickering rate, signal peptide sensitivity or pore size may be obtained depending upon these factors. This should be mentioned, and in any case taken into account whenever comparing electrophysiological results, and drawing conclusions. Also, when possible try to refer to those studies closest to native conditions.
9) In line 226 it is stated that “The TOB/SAM complex was also suggested to be involved in apoptosis” and refs. 73 and 74 are given to support this statement. However, ref. 74 describes the discovery of MAC. Equally, ref. 73 does not deal with the SAM complex.
10) Lines 227-229: “An 226 important study on yeast and human cells conducted by [70] revealed that Tob55/Sam50 227 was required for the transport of granzymes (a and b) and caspase-3 to cross the outer 228 membrane” However, ref. 70 deals with how expression levels in yeast of some mitochondrial proteins (among them Sam50) is influenced by VDAC.
11) Line 247. A reference should be given to support that Mdm10 mediates the interaction between SAM and ERMES.
12) Line 250-251, ref. 68 does not give the diameter of the Sam50 channel. Also, what is central density?
13) Line 255-256: “Tob55/Sam50a is responsible for channel formation, while isoform b is responsible for releasing incoming precursor proteins from the channel [89]”. Ref. 89 is about VDAC.
14) Lines 261-263. VDAC voltage dependence is complicated, since it depends on the system and technique. Please revise.
15) Line 424: “The observed cation selectivity of the Ayr1 channel suggests its contribution to protein import” The statement is too strong. Cation selectivity is not enough to suggest protein import features. Please moderate the statement.
16) Several typo errors are scattered throughout the text, some of which are listed below:
Figure legend 1: (MICOS) complex complex… Repetition.
Line 187: conductive states should be conductance states.
Line 192: channel conductivity should be channel conductance.
Line 610: Kinnallyet, K.W. should be Kinnally, K.W.
Line 301: in-fluences should be influences.
Line 302: do-main should be domain.
Line 432: cy -tosol should be cytosol
Line 682: translocationassociated should be translocation associated.
Lines 177 and 413: S. cerevisiae should be S. cerevisiae
Author Response
Mitochondrial channels play numerous and most probably yet undefined functions, decisive for cell’s normal behavior. This is a well written and focussed review that summarizes some studies describing the most outstanding structural and functional features of the large channels and protein import related complexes found in the mitochondrial outer membrane.
This is a good, clearly presented and up to date revision that deserves to be published, as it has the potential to interest a wide audience, including scientists not clearly involved in the field. However, a number of changes should be considered to improve the quality and accuracy of the manuscript:
We are very grateful for all the comments allowing improvement of our manuscript.
- According to the title, membrane permeability modulation is one of the prominent aspects of this manuscript. However, this issue is merely scattered, and thus blurred throughout the text. More emphasis or a devoted section to this growing up theme might be considered.
- We agree with the Reviewer that the membrane permeability modulation is merely scattered in the text, however it was not our intension to focus on the membrane modulation. Our aim was rather to present the wide variety of targets for possible modulation. Therefore we have changed the title of the manuscript.
From „The Diversity of Mitochondrial Outer Membrane Protein Import Channels: Emerging Possibilities in Membrane Permeability Modulation”
to „The Diversity of the Mitochondrial Outer Membrane Protein Import Channels: Emerging Targets for Modulation”
We do hope the new title is more relevant to the review content.
- Although the information is exact, there is a weakness that should be underlined. The authors do not cover any aspects, electrophysiological nor biochemical, of the Mitochondrial Apoptosis-induced Channel (MAC), a large protein transport channel also identified in the outer membrane.
The Reviewer is right. We did not include MAC as we concentrated on channels involved in protein import. However, taking the Reviewer comment into account, we added a new section devoted to the channel (line 366). The section (3.5) is entitled Mitochondrial Apoptosis-induced Channel (MAC). The content of the section is as follows:
MAC is the outer membrane large channel known as an early marker of the onset of apoptosis. It participates in release of proteins normally constrained within the intermembrane space, such as cytochrome c, second mitochondria – derived activator of caspases (Smac)/Direct inhibitor of apoptosis-binding protein with low pI (DIABLO) or apoptosis induced factor (AIF) [75,105]. MAC is formed by Bax and/or Bak proteins, and at least one of the proteins must be present. Bax protein is cytosolic whereas Bak is an integral protein of the outer membrane and they both stay in an inactive form until MAC formation [75,105]. The activity of MAC is significantly different from the TOM and TOB/SAM complexes. Moreover the former are constitutive channels of the outer membrane whereas MAC activity co-occur only with apoptosis [75,76,105]. MAC activity is regulated by other Bcl-2 family proteins. Bcl-2 and Bcl-xL inhibit apoptosis by separating Bax and Bak. Moreover, MAC can be block by dibucaine, trifluoperazine and propranolol in a dose-dependent manner) as well as 3.6-dibromocarbazole piperazine derivatives of 2-propanol (that blocked cytochrome c release induced in isolated mitochondria by tBid).
By application of patch-clump, the reconstituted MAC conductance was estimated as heterogenous; i.e between 1.5–5 nS (1500-5000 pS in 150 mM KCl). MAC frequently fluctuates between the fully open state and fully closed state, with a maximal single transition size of 2000 pS in 150 mM KCl and at least three substates [75]. MAC is a voltage-independent channel and is slightly cation-selective. As it has been shown for mammalian apoptotic FL5.12 cells after interleukin 3 (IL-3) withdrawal, PK+/PCl- = 3:1. The cation selectivity is consistent with its putative role in releasing cationic proteins such as cytochrome c [75]. At high conductance state MAC is permeable dextran of molecular weight in the range of 10 to 17 kDa, but not in the range of 45 to 71 kDa. Based on the polymer exclusion method, MAC pore diameters was estimated in the range of 2.9 to 7.6 nm [75,76].”
We have also added the data concerning MAC into the Table 1
- The channel of the TOM complex is reported as the dimeric or tretrameric forms of Tom40. However, the trimeric form, its dynamics and function are ignored (e.g. Shiota, T. et al. Molecular architecture of the active mitochondrial protein gate. 2015. Science 349: 1544–1548. doi:10.1126/science.aac6428).
We agree with the Reviewer. We added the following information to the section 3.1:
-line 137-144:
„In the previous study [47,48], two kinds of the TOM complex were observed: three-channel complex representing trimeric form of Tom40 and two-channel complex representing dimeric form of Tom40. It was showed that the two channel complex consisted of two Tom40 and small Tom subunits but did not contain Tom22. It was concluded that the two channel complex dimmer form functioned as a premature complex [49] whereas the mature TOM complex dynamically exchanges with the two channel one providing an assembly platform for the integration of new subunits”.
- In line with the above, evidence for structurally distinct Tom40 conformations is mentioned (e.g. ref. 54), but their dynamic implications in substrate recognition and in transport function are largely omitted.
We agree with the Reviewer, so we added the issue to the manuscript (section 3.1; lines 174-179):
„This results demonstrated that all studied Tom40 constructs formed channels of different conductance. Each of the expressed proteins interacted with a pre-sequence peptide in the concentration dependent manner although the interaction with the conserved N - terminal Tom40 domain was not required. This study showed that Tom40 dimers were functional and their distinct conformations were regulated by binding of the substrate”.
- In general, bibliographic references are timely and well incorporated into the review. However, citations concerning functional aspects (namely the electrophysiology) are scarce in relation to those referring to structural aspects.
Indeed, this results from the narrow branch of the study. There are not many groups working on this issue.
- The electrophysiology of Tom40 is confusing. E.g. the way to report the size of the channel in different organisms is tangled by the different salt concentrations used in electrophysiological approaches and misleads the non-expert readers. Too many distracting details sidetrack the reader from the main message, i.e. a similar conductance size.
We agree with the Reviewer. In the Table 1 we corrected the value of KCl concentration and channel conductance presented in pS except of the data concerning MAC where it was added that the data was obtained by different method - patch clump. We introduced the following corrections:
- line 209-211:
The sentence: ”After extrapolation, this result is in agreement with the data obtained for S. cerevisiae and N. crassa.”
was changed into: „After extrapolation (625 pS and 575 pS respectively in 250 mM KCl), this result is in agreement with the data obtained for S. cerevisiae and N. crassa.”
- line 297-299:
The sentence: „This result is in agreement with data obtained by [8] where (by extrapolation) Tob55/Sam50 showed a channel activity level of 3.75 nS in 1M KCl”.
was changed into: „This result is in agreement with data obtained by [8] where Tob55/Sam50 showed a channel activity level of 3.75 nS in 1M KCl what after extrapolation correspond to 925 pS in 250 mM KCl”.
- line 504-506:
The sentence: ”Ayr1 reconstituted in liposomes or planar lipid bilayers revealed a main conductance of 1.47nS in the fully open state”.
was changed into: „Ayr1 reconstituted in liposomes or planar lipid bilayer membrane revealed a main conductance of 1.47 nS in 1 M KCl (367 pS at 250 m M KCl after extrapolation) in the fully open state”.
The data in the Table 1 were also changed properly.
Table 1 does not settle much the issue of significance or relevance of this type of studies, as conductance, reversal potential or permeability ratios are not discussed nor interpreted in light of their possible functional significance. By the way, at this level, reversal potential and ion permeability ratios, basically provide the same information. Whereas other relevant channel properties (e.g. voltage dependence, signal peptide sensitivity, number of substates etc.) are omitted. Also on this table, differences between transported molecules and modulators of the channel activity is not too precise. For instance, Aβ is under the modulator column, whereas ref. 67 points to it as transported molecule. Finally, the heading of this table is striking. I do not recall any mitochondrial outer membrane channels experiments performed “in vivo”.
The missing phrases were added in the case of TOM and TOB/SAM complexes as well as Ayr1 protein.
- section 3.1 line 132-133:
“Moreover it was also proposed that the TOM complex can act as an insertase mediating lateral release of the membrane imported proteins into the outer membrane [44]”.
-line 196-201
„Tom40 is the voltage dependent channel. In the absence of membrane potential, the reconstituted channel is completely open but closes symmetrically at positive or negative potentials. The studies on the reconstituted S. cerevisiae Tom40 showed that in the presence of low membrane potential the pre-sequence peptide selectively bound to Tom40 and thereby altered the gating of the channel but was rapidly translocated through the channel when driven by higher voltage [40]”.
-section Tob55/Sam50.
The following correction were introduced:
-line 303-305: the sentence was changed as follows: „It has also been shown that the channel opening is mediated by a precursor protein β-signal that causes the displacement of the endogenous carboxy-terminal β-signal of the Tob55/Sam50 channel [95]”
- section 4.3 line 511-513:
“The presented parameters of the Ayr1 could be consistent with the protein localization in the mitochondrial outer membrane and ER fractions, as being arranged in contact sites between ER and mitochondria”.
We are sorry, the amyloid β peptide should be presented as a transported molecule. For clarity, we changed Table 1 to present modulators and transported molecules more precisely. We also removed the heading „in vivo” experiments.
- Electrophysiological studies should bring for more than just a list of physical characteristics of the channels. Functional implications of these characteristics are lacking and must be outlined (e.g. what is the relationship between the size or the fully open and the half open states of the channel, and the size of a protein in transit through the channel?
This information was enclosed in the part concerning MAC (see above our answer to your comment no 2). Unfortunately in the case of the other channels the data are not available.
Are the channels always open? What triggers the gating of the channels? Besides, all these are important aspects related to membrane permeability regulation.
This information was added to the Introduction
-line 41-44
“Protein channels described in this review except of the MAC are constitutively open and do not need activators to regulate its opening. This phenomena makes them different than channels located within the inner mitochondria membrane”.
We have also added the new column into the Table 1 containing the relevant data.
In line with some inaccuracies found in the electrophysiology, the statement about flickering (lines 189-191), as it is (i.e. without further interpretation) is irrelevant.
-Section 3.1 line 216-217
We added the following phrase:
“This data indicated flickering which is typical for this kind of the outer membrane channels”.
Statement on lines 192-193 is also misleading: Signal peptides do not measure the regulation of channel conductance (rather than conductivity). Signal peptides trigger the channel activity by increasing its flickering, i.e. rapid transitions from the fully open to the substate and close state. Even more, ref. 58 does not refer to this aspect but rather to the recognition of preproteins by the isolated TOM complex. No mention is given to whether the channel remains or not closed in the absence of signal peptides/preproteins and if signal peptides/preproteins are the ones triggering the opening of the channel.
We agree with the Reviewer and changed the sentence as follows:
- Section 3.1 line 218
From: “The regulation of channel conductivity was measured using the specific signal peptides known to be recognised by the TOM complex receptors [58].”
To:
„The channel conductance was measured using the specific signal peptides known to be recognised by the TOM complex receptors [61]. The presence of signal peptides resulted in the channel flickering, i.e. rapid transitions from the fully open state to substates of lower conductance resulting from the channel blockade. This is a consequence of signal peptides interaction with TOM complex subunits”.
- It is well known than the reconstituted system (i.e. isolated protein, purified complex, proteoliposomes, or native mitochondrial membranes) as well as the electrophysiological technique applied (i.e. planar bilayer vs patch-clamp) greatly influence the outcome of the channel properties obtained. Thus, dramatic differences in voltage dependence, ion selectivity, reversal potential, flickering rate, signal peptide sensitivity or pore size may be obtained depending upon these factors. This should be mentioned, and in any case taken into account whenever comparing electrophysiological results, and drawing conclusions. Also, when possible try to refer to those studies closest to native conditions.
We agree with the Reviewer and added the following sentences concerning the issue:
- Section 3.1 line 240-245:
„It should be mentioned that the means of protein preparation for the studies on Tom40 and other described in this review protein channels as well as the applied methods as planar lipid bilayer membranes and patch clump could greatly influence the determined channel properties. Thus, combining electrophysiological data with structural data provide reasonable background for putative modulation. However, this always would need verification under cellular conditions.”
- In line 226 it is stated that “The TOB/SAM complex was also suggested to be involved in apoptosis”and refs. 73 and 74 are given to support this statement. However, ref. 74 describes the discovery of MAC. Equally, ref. 73 does not deal with the SAM complex.
We are sorry for this mistake. It was corrected to [72] in line 265.
- Lines 227-229: “An 226 important study on yeast and human cells conducted by [70] revealed that Tob55/Sam50 227 was required for the transport of granzymes (a and b) and caspase-3 to cross the outer 228 membrane”However, ref. 70 deals with how expression levels in yeast of some mitochondrial proteins (among them Sam50) is influenced by VDAC.
We are sorry for this mistake. It was corrected to [72] in line 266.
- Line 247. A reference should be given to support that Mdm10 mediates the interaction between SAM and ERMES.
We agree with the Reviewer and we added [83,91,92] in line 288.
- Line 250-251, ref. 68 does not give the diameter of the Sam50 channel. Also, what is central density?
We are sorry, It’s a mistake: it should be “central cavity” – it was corrected. And correct ref. is [3,11] in line 290.
13) Line 255-256: “Tob55/Sam50a is responsible for channel formation, while isoform b is responsible for releasing incoming precursor proteins from the channel [89]”. Ref. 89 is about VDAC.
We are sorry for this mistake. It was corrected ([93] in line 295).
- Lines 261-263. VDAC voltage dependence is complicated, since it depends on the system and technique. Please revise.
The sentence was corrected as follows:
- Section 3.2 line 300-303
From: „ At voltages greater than +/- 70 mV, the channel formed by Tob55/Sam50 is partially closed, which clearly distinguishes this channel from the VDAC channel [89]”.
To: „At voltages greater than +/- 70 mV, the channel formed by Tob55/Sam50 is partially closed, which clearly distinguishes this channel from the VDAC channel measured by the same technique as planar lipid bilayers [94]”.
- Line 424: “The observed cation selectivity of the Ayr1 channel suggests its contribution to protein import”The statement is too strong. Cation selectivity is not enough to suggest protein import features. Please moderate the statement.
We agree with the Reviewer. The sentence was corrected as follows:
- Section 3 line 510-511:
From: „The observed cation selectivity of the Ayr1 channel suggests its contribution to protein import”
To: “The observed cation selectivity of the Ayr1 channel might suggest its contribution to protein import”.
16) Several typo errors are scattered throughout the text, some of which are listed below:
Figure legend 1: (MICOS) complex complex… Repetition.
Line 187: conductive states should be conductance states.
Line 192: channel conductivity should be channel conductance.
Line 610: Kinnallyet, K.W. should be Kinnally, K.W.
Line 301: in-fluences should be influences.
Line 302: do-main should be domain.
Line 432: cy -tosol should be cytosol
Line 682: translocationassociated should be translocation associated.
Lines 177 and 413: S. cerevisiae should be S. cerevisiae
We thank the Reviewer for the typo error indication. All the errors were corrected.
Reviewer 3 Report
In the review "The Diversity of Mitochondrial Outer Membrane Protein Import Channels: Emerging Possibilities in Membrane Permeability Modulation" (molecules-1269124), Mazur et al. give an overview of the current knowledge of the channels mainly employed for protein transport across the outer membrane of mitochondria (OMM). It seems that the authors have made a lot of effort to include all available measured properties of the OMM complexes in the manuscript, however, they are somewhat missing out on making sense of these parameters in a physiological context. The manuscript is recommended for publication in Molecules given that the below details are dealt with.
Major comments
The Title, Abstract, Introduction and Conclusions mention "channel modulation" and "modules" repeatedly, however the text gives very few examples of channel activity modulators. The authors should give clear definitions of these expressions, clarify their physiological relevance/implications and develop this concept better.
In the present version of the manuscript the ion conductance measurements on single OMM complexes only seem to be physical parameters (along with the dimensions of the complexes) with quite limited biological value. For the benefit of the general reader, the authors should explain what these measurements mean for the physiological role of the complexes. Also a comparative analysis of the measurements in a physiological context would be in place given that the complexes appear all together in the OMM in vivo.
The authors should be invited to extend their description of the molecules, besides proteins, transported by the OMM complexes and how they achieve their specificity.
Section 4.2 "Transport of metabolites". The authors should describe better what are the characteristics of the molecules (size, charge, examples) that may cross the outer membrane and those of the compounds that may not pass.
Minor comments
Quite many hyphenation typos are found throughout the manuscript.
Line 39 says "OM7". It should be "OMC7".
Line 312-313 phrase "The expressed..." should be reformulated to make sense.
Table 1 is unnecessary complicated and not aesthetically constructed. The lines should be clearly separated/indicated to read the data in horizontal so that protein name is connected to conductance, organism and reference (especially for Tom40).
Author Response
In the review "The Diversity of Mitochondrial Outer Membrane Protein Import Channels: Emerging Possibilities in Membrane Permeability Modulation" (molecules-1269124), Mazur et al. give an overview of the current knowledge of the channels mainly employed for protein transport across the outer membrane of mitochondria (OMM). It seems that the authors have made a lot of effort to include all available measured properties of the OMM complexes in the manuscript, however, they are somewhat missing out on making sense of these parameters in a physiological context. The manuscript is recommended for publication in Molecules given that the below details are dealt with.
- We would like to thank the Reviewer for detailed revision. We would like to explain that our intention was to present the wide variety of targets for modulation. Therefore, for clarity, we propose change of the title. We do hope that the new title is more relevant to the review content. So the title was changed as follows:
From: „The Diversity of Mitochondrial Outer Membrane Protein Import Channels: Emerging Possibilities in Membrane Permeability Modulation”
To:„ The Diversity of Mitochondrial Outer Membrane Protein Import Channels: Emerging Target for Modulation”
Major comments
The Title, Abstract, Introduction and Conclusions mention "channel modulation" and "modules" repeatedly, however the text gives very few examples of channel activity modulators. The authors should give clear definitions of these expressions, clarify their physiological relevance/implications and develop this concept better.
We agree with the Reviewer. We have changed the statement as follows:
Section Abstract, line 12-16:
The sentence:” To date, eight channels have been identified in the mitochondrial outer membrane—of which at least half represent a mitochondrial protein import apparatus. Nevertheless, these channels appear to be flexible in their specificity and amenable to modulation. In this review, we focus on the channel structure, properties and transported molecules as well as aspects important to channel modulation.”
was changed into: “To date, nine channels have been identified in the mitochondrial outer membrane—of which at least half represent a mitochondrial protein import apparatus. When compared to the mitochondrial inner membrane, the presented channels are mostly constitutively open and consequently may participate and transport of different molecules and contribute to relevant changes in the outer membrane permeability and the channel conductance”.
-Section Conclusions, line 524-531:
the sentence: “Altogether, this could be a subject of sophisticated modulation that is important for mitochondrial functions. Moreover, this indicates that the mitochondrial outer membrane is a compartment that has not yet been fully described. Undoubtedly, these results highlight many implications for channel modulation and the treatment of mitochondria-linked diseases.
was changed into: “Undoubtedly, these results highlight many implications for channel modulation that might affect mitochondrial protein turnover as well as transport of different small molecules including crucial metabolites as ATP, ADP, substrates of the respiratory chain and superoxide anion as well as different drugs as granzymes. This in turn might be beneficial for treatment of mitochondria-linked diseases mediated by the outer membrane permeability impairment as cancer by inducing apoptosis [75] as well as by incorrect intramitochondrial processes (e.g. neurodegenerative disorders as Parkinson and Alzheimer diseases).”
No mention is given to whether the channel remains or not closed in the absence of signal peptides/preproteins and if signal peptides/preproteins are the ones triggering the opening of the channel.
-Section Introduction line 47-50
The sentence: „Such knowledge is crucial for applying these channels as modulation targets, which could affect the cellular processes mediated by the channels as well as mitochondrial functions”.
Was changed into: „Such knowledge is crucial for development of the channels’ modulators which could prove effective in correcting mitochondrial processes that in turn may constitute an important therapeutic approach in disease treatment.”
In the present version of the manuscript the ion conductance measurements on single OMM complexes only seem to be physical parameters (along with the dimensions of the complexes) with quite limited biological value. For the benefit of the general reader, the authors should explain what these measurements mean for the physiological role of the complexes. Also a comparative analysis of the measurements in a physiological context would be in place given that the complexes appear all together in the OMM in vivo.
We agree with the Reviewer. Therefore we introduced the following corrections:
Section Introduction, line 41—44:
The sentence was added: „Protein channels described in this review except of the MAC are constitutively open and do not need activators to regulate its opening. This phenomena makes them different than channels located within the inner mitochondria membrane”.
The example of this point can be found in the new section (3.5) entitled Mitochondrial apoptosis channel (MAC) channel:
-Section 3.5, line 366-391
MAC is the outer membrane large channel known as an early marker of the onset of apoptosis. It participates in release of proteins normally constrained within the intermembrane space, such as cytochrome c, second mitochondria – derived activator of caspases (Smac)/Direct inhibitor of apoptosis-binding protein with low pI (DIABLO) or apoptosis induced factor (AIF) [75,105]. MAC is formed by Bax and/or Bak proteins, and at least one of the proteins must be present. Bax protein is cytosolic whereas Bak is an integral protein of the outer membrane and they both stay in an inactive form until MAC formation [75,105]. The activity of MAC is significantly different from the TOM and TOB/SAM complexes. Moreover the former are constitutive channels of the outer membrane whereas MAC activity co-occur only with apoptosis [75,76,105]. MAC activity is regulated by other Bcl-2 family proteins. Bcl-2 and Bcl-xL inhibit apoptosis by separating Bax and Bak. Moreover, MAC can be block by dibucaine, trifluoperazine and propranolol in a dose-dependent manner) as well as 3.6-dibromocarbazole piperazine derivatives of 2-propanol (that blocked cytochrome c release induced in isolated mitochondria by tBid).
By application of patch-clump, the reconstituted MAC conductance was estimated as heterogenous; i.e between 1.5–5 nS (1500-5000 pS in 150 mM KCl). MAC frequently fluctuates between the fully open state and fully closed state, with a maximal single transition size of 2000 pS in 150 mM KCl and at least three substates [75]. MAC is a voltage-independent channel and is slightly cation-selective. As it has been shown for mammalian apoptotic FL5.12 cells after interleukin 3 (IL-3) withdrawal, PK+/PCl- = 3:1. The cation selectivity is consistent with its putative role in releasing cationic proteins such as cytochrome c [75]. At high conductance state MAC is permeable dextran of molecular weight in the range of 10 to 17 kDa, but not in the range of 45 to 71 kDa. Based on the polymer exclusion method, MAC pore diameters was estimated in the range of 2.9 to 7.6 nm [75,76].”
Also a comparative analysis of the measurements in a physiological context would be in place given that the complexes appear all together in the OMM in vivo.
The comparative analysis is presented in the revised Table 1. For clarity, we changed the way of presenting modulators and transported molecules as well as added a column indicating whether a given channel is constitutively open or not.
The authors should be invited to extend their description of the molecules, besides proteins, transported by the OMM complexes and how they achieve their specificity.
Section 4.2 "Transport of metabolites". The authors should describe better what are the characteristics of the molecules (size, charge, examples) that may cross the outer membrane and those of the compounds that may not pass.
The section was corrected accordingly.
-line 450-456:
“The molecular mass cut-off for the transported molecules is assumed to be about 4 kDa although the limit could be decreased when VDAC switches to lower conducting substates featuring less anion selectivity [132]. Nevertheless, VDAC is considered to be the major transport pathway for compounds as diverse as inorganic ions (e.g., K+, Na+ and Cl-), metabolites of different size and charge (e.g., big anions as ATP, AMP and glutamate, and small anions as superoxide anion as well as big cations as NADH and acetylocholine) and large macromolecules such as tRNA (e.g. [129,133,134]).”
-line 460-468:
“This assumption was initially based on observation that in the absence of yVDAC1, NADH, ADP and CATR (carboxyatractylate) displayed limited access into the mitochondrial intermembrane space and the observed limitations depended on charge and size of these molecules (the highest for CATR being big anion of molecular weight higher than ADP and the lowest for NADH being big cation of molecular weight slightly smaller than CATR). Moreover, the limitations were weakened by the presence of Mg2+ that together implies involvement of cation selective channel of lower conductance that yVDAC1 [135]. This in turn correlates with the electrophysiological characteristic of the TOM complex channel (see p. 3.1 for details).”
Minor comments
Quite many hyphenation typos are found throughout the manuscript.
Line 39 says "OM7". It should be "OMC7".
- It was corrected
Line 312-313 phrase "The expressed..." should be reformulated to make sense.
-line 353-354:
The phrase was corrected as follows: “The S. cerevisiae Mim1, expressed in E.coli and then purified and renatured, was studied in planar lipid bilayer membranes”
Table 1 is unnecessary complicated and not aesthetically constructed. The lines should be clearly separated/indicated to read the data in horizontal so that protein name is connected to conductance, organism and reference (especially for Tom40).
We thank the Reviewer. It was corrected. We hope it`s more clear at the revised form.